# DYNAMIC MEMORY BASED ADAPTIVE OPTIMIZATION

## ABSTRACT

Define an optimizer as having memory $k$ if it stores $k$ dynamically changing vectors in the parameter space. Classical SGD has memory 0, momentum SGD optimizer has 1 and Adam optimizer has 2. We address the following questions: *How can optimizers make use of more memory units? What information should be stored in them? How to use them for the learning steps?* As an approach to the last question, we introduce a general method called "Retrospective Learning Law Correction" or shortly RLLC. This method is designed to calculate a dynamically varying linear combination (called *learning law*) of memory units, which themselves may evolve arbitrarily. We demonstrate RLLC on optimizers whose memory units have linear update rules and small memory ($\leq 4$ memory units). Our experiments show that in a variety of standard problems, these optimizers outperform the above mentioned three classical optimizers. We conclude that RLLC is a promising framework for boosting the performance of known optimizers by adding more memory units and by making them more adaptive.

## 1 INTRODUCTION

In this paper, we investigate optimizers that store $k$ vectors in the parameter space $\mathbb{R}^n$ of a neural network or more generally in the parameter space related to any optimization problem. We call such vectors *memory units* and we measure the memory usage of an optimizer by the number them.

The simplest example for an optimizer with memory is the momentum SGD optimizer which stores a single vector $m$ (*momentum vector*) in the parameter space $\mathbb{R}^n$. In each step, $m$ is updated according to the rule $m \longleftarrow \beta m + \bigtriangledown_\theta f(\theta)$ where $f$ is the objective function, $\theta \in \mathbb{R}^n$ is the parameter vector and $0 < \beta < 1$ is a fixed real number. The vector $\theta$ is updated according to the rule $\theta \longleftarrow \theta - cm$ where $c > 0$ is the learning rate.

The Adam optimizer operates with two memory units. One of them is the *momentum vector* and the other one is the momentum of the squares of the gradient vectors. In contrast with the momentum optimizer, the Adam optimizer is not linear in the gradient vectors. Neither the update rule of the memory units, nor the way the memory units are used for the parameter update is linear.

The present paper has two independent contributions. The first contribution is a novel and simple method that we call RLLC=*Retrospective Learning Law Correction*. It is an update rule for a vector $L$ (called *learning law*) that describes a natural way of using a set of dynamically changing memory units for the update of the parameter vector $\theta$. More precisely, $L \in \mathbb{R}^k$ contains the coefficients of a linear combination of the $k$ memory units which is multiplied by a fixed learning rate $c_1$ and substracted from $\theta$ as usual. In each step, before updating $\theta$ and the memory units, we update $L$ by the formula $L \longleftarrow L + c_2 M^+ g$ where $M$ is the $n \times k$ matrix formed by the memory units, $M^+$ is the Moore-Penrose inverse of $M$, $g$ is the newly received gradient and $c_2$ is the *meta learning rate*.

Note that in practice we use the formula $M^+ = B^+ M^T$ for calculating $M^+$, where $B = M^T M$ is the so-called Gram matrix of the memory units. Since $B$ is a $k \times k$ matrix, where $k$ is a small number (at most 4 in our examples), the calculation of $B^+$ is a negligible part of the computational load. This means that the computational cost of the RLLC step comes mostly from calculating the matrix products $M^T M$ and $B^+ M^T$, which involves only a few elementary operations per parameter if $k$ is small. Our experiments (see Appendix A.4.2) show that RLLC optimizers with small memory have a runtime similar to that of more classical optimizers, such as Adam.

The main idea behind the RLLC update rule for $L$ is that the new gradient $g$ contains retrospective information on how the algorithm could have performed better in the previous step. Thus it can be used to compute a corrected version of $L$ which "thinks more ahead". Note that our update rule of the learning law can also be regarded as a general framework for associating a $k$-dimensional adaptive learning rate with an arbitrary set of $k$ evolving memory units.

As the second main contribution, we examine optimizers in which memory units are updated by fixed linear rules. More precisely, in each step each memory unit is updated to a linear combination of the memory units and the new arriving gradient. The parameter vector is updated by a (possibly changing) liner combination (given by the learning law $L$) of the memory units. Such optimizers are interesting even if the learning law is fixed. They include SGD, momentum SGD and Nesterov Accelerated Gradient (NAG) Nesterov (2012). Thus, the linear framework provides a useful generalization of these famous optimizers and enables a dynamically changing continuous interpolation between them. The RLLC method turns out to be ideal for this. Our experiments show that linear memory combined with RLLC leads to powerful optimizers. The case of memory 1 is already interesting. A memory 1 linear optimizer stores a momentum vector. Applying RLLC in this trivial setting yields a variant of the momentum SGD optimizer enhanced with a new type of adaptive learning rate. As the number of memory units increases, the mathematics becomes more complex, presenting a field of study that is interesting in its own right. We present some of the fundamental properties of linearly updated memory units. In particular, we prove a version of basis independence for RLLC combined with linear memory which allows us to apply basis transformations to the update rules without changing the optimization process. This together with a variant of the Jordan normal form over the field $\mathbb{R}$ helps to convert these optimizers into a canonical form in which each memory unit is associated with a so-called Jordan block. A Jordan block of size 1 corresponds to a single memory unit (denoted by $M(\beta)$) storing a momentum vector of the gradients with parameter $\beta$. A Jordan block of size 2 either corresponds to a pair of memory units (denoted by $CM(\beta), \beta \in \mathbb{C}$) namely the real and imaginary parts of a momentum vector with complex parameter or to a pair of memory units $m_1, m_2$ (denoted by $M_2(\beta)$) where $m_1$ is a momentum vector of the gradient and $m_2$ is a momentum vector of $m_1$, both with parameter $\beta$. In general, there are two infinite families of Jordan blocks giving rise to $k$-tuples or $2k$-tuples of memory units denoted by $M_k(\beta)$ and $CM_k(\gamma)$. These are the fundamental building blocks of linearly updated memory. We denote the natural operation by $\oplus$ which combines these building blocks into larger memory by the union of the corresponding memory units. By slightly abusing the notation we often identify memory update rules with optimizers where learning is given by the RLLC method. For example, $M(\beta)$ also denotes the memory 1 optimizer with memory unit $M(\beta)$ (a momentum vector) and with RLLC. Notice that the $M(\beta)$ optimizer is a close relative of momentum SGD but it is not equivalent with it.

In our experiments, we identified a number of interesting simple settings involving few (at most 4) memory units. These include the types of optimizers $M(\beta)$, $M(\beta) \oplus M(0)$, $M(\beta_1) \oplus M(\beta_2) \oplus M(\beta_3)$, $M_2(\beta)$, $M_3(\beta)$ and $M(\beta) \oplus M(-\beta) \oplus CM(\beta i)$. We observed that these optimizers often surpassed the performance of three commonly used optimizers across a variety of tasks even without carefully optimizing the parameters $\beta_i$. Notice that $M(\beta) \oplus M(0)$ is an adaptively changing linear combination of SGD, momentum SGD and NAG. Thus, it adaptively interpolates between three well known optimizers (for details see appendix). Remarkably, it demonstrated competitive or even superior performance compared to the Adam optimizer in many tasks, which also uses two memory units.

This paper primarily aims not to challenge all existing optimizers in the field, but rather to introduce a novel mathematical concept that could spark further research. The experimental results presented here should be interpreted as an illustration of the potential of our approach. We posit that the implications of the RLLC method extend beyond mere enhancements to current optimization techniques, suggesting broader applications and insights in the realm of optimization and machine learning.

## 2 RELATED WORK

Similar to our RLLC method and our framework of linear optimizers, various other optimizers (including classical ones such as NAG Nesterov (2012) and Adam Kingma & Ba (2014)) are based on storing vectors in the parameter space to enhance performance. A more recent work in this area is

McRae et al. (2022), titled "Memory Augmented Optimizers for Deep Learning." The main novelty of this paper is its method for selecting, storing, and using a list of 'critical gradients.'

Our method has a close connection to adaptive learning rate methods Wang et al. (2022); Keskar & Socher (2017b), as highlighted by the fact that the RLLC method used for a single memory unit is equivalent to a new type of adaptive learning rate. Other notable papers in this area include the following: AdaBound Luo et al. (2019), which combines the benefits of adaptive methods and SGD by dynamically bounding the learning rates during training, aiming for fast convergence and improved generalization; AdaBelief Zhuang et al. (2020), which adapts step sizes based on the "belief" in observed gradients by comparing them to an exponential moving average of past gradients, enhancing both convergence speed and generalization; and DiffGrad Dubey et al. (2020), which adjusts learning rates based on the differences between the current and immediate past gradients, emphasizing updates where the gradient changes rapidly.

Our work is also related to the broader field of metaoptimization, where the optimization methods themselves are optimized, often through learned strategies. In this context, the main approach is training an auxiliary model, often a neural network, to optimize the primary model tasked with solving the original problem. Some of these methods leverage traditional gradient-based optimizers (Andrychowicz et al., 2016; Metz et al., 2022b; Bengio et al., 1991; Wichrowska et al., 2017), while others explore alternative methods such as evolutionary algorithms (Bengio et al., 1991; Metz et al., 2020) or reinforcement learning strategies (Li & Malik, 2017; Bello et al., 2017). The goal of these methods is to design optimizers through learned processes, which can outperform standard, hand-designed optimizers across a wide variety of tasks.

Finally we mention a recent and independent paper Pagliardini et al. (2024), which introduces an update to the Adam optimizer by replacing its momentum vector with a (typically fixed) linear combination of two momentum vectors with different decay parameters. The idea of combining multiple momentum vectors with different decay parameters is also an important idea in our paper. However, the main novelty of our work lies in the new methodology, called the RLLC method, which adaptively adjusts the linear combination (learning law) of the momentum vectors (or any other useful vectors). Our RLLC method is essentially a gradient descent algorithm (with a dynamically changing objective function) in the space of potential learning laws, utilizing knowledge from the previous training step. This meta-learning style adaptive nature is one of the key differences between our work and the previously mentioned papers.

## 3 RETROSPECTIVE LEARNING LAW CORRECTION

**Functional approach to optimizers:** The RLLC method is presented through an abstract mathematical framework for optimizers. This framework is somewhat specialized, yet it maintains sufficient generality to encompass a range of interesting optimizers. We think of optimizers as entities with an evolving internal state that updates at each step based on newly received gradients. Additionally, the optimizer calculates a parameter update vector relevant to the optimization process. A functional description of such an optimizer is given in the following definition.

**Definition 3.1.** An *optimizer* for $n$ parameters is a pair of functions of the form $F : \mathcal{S} \times \mathbb{R}^n \to \mathcal{S}$ and $G : \mathcal{S} \times \mathbb{R}^n \to \mathbb{R}^n$ where $\mathcal{S}$ is the set of possible internal states, $F$ is the *state update function* and $G$ is the *parameter update function*.

To translate optimizers into an actual optimization process we choose an initial internal state $S_0 \in \mathcal{S}$ and an initial parameter vector $\theta_0 \in \mathbb{R}^n$. Then we iterate

$$S_t := F(S_{t-1}, g_t), \ \ \theta_t := \theta_{t-1} + G(S_t, g_t)$$

where $g_t$ is a gradient vector received by the optimizer in the $t$-th step. To illustrate this formalism, assume that the optimizer is given by $\mathcal{S} = \mathbb{R}^n$, $F(v, w) = \beta v + w$, $G(v, w) = -cv$. In this case, we obtain the momentum SGD with learning rate $c$ and decay parameter $\beta$.

**Optimizers with memory and RLLC:** We will think of memory $k$ optimizers in a way that the internal state space is of the form $\mathbb{R}^{n \times k} \times \mathcal{H}$ where the columns of matrices in $\mathbb{R}^{n \times k}$ represent $k$ vectors in the parameter space $\mathbb{R}^n$ and $\mathcal{H}$ will be called the space of hidden states. We typically assume that $n$ is a large number and that the hidden states are described by much fewer than $n$ parameters. A *memory update rule* for $k$ memory units is a function of the form $U : (\mathbb{R}^{n \times k} \times \mathcal{H}) \times \mathbb{R}^n \to \mathbb{R}^{n \times k} \times \mathcal{H}$

where the external $\mathbb{R}^n$ component represents new arriving gradients. Such a function does not yet determine an optimizer. The RLLC method is designed to turn memory update rules into optimizers by extending their state space with a vector called learning law and introducing a natural parameter update function. We give two different descriptions of RLLC. The first one is a functional description which is more convenient for proofs.

We will need the so-called Moore-Penrose inverse which is defined for an arbitrary matrix $A \in \mathbb{R}^{n \times k}$ and is denoted by $A^+$. Note that if $A$ has rank $k$ (which means that $A$ is non-degenerate if $n \geq k$) then $A^+ = (A^T A)^{-1} A^T$.

**Definition 3.2** (RLLC functional form). Let $U$ be a memory update rule as above. Then the corresponding RLLC optimizer with learning rates $c_1, c_2$ is given as follows. The state space is $\mathcal{S} := \mathbb{R}^{n \times k} \times \mathcal{H} \times \mathbb{R}^k$ where the extra component $\mathbb{R}^k$ is called the learning law. The functions $F, G$ are given in the following way. Assume that $M \in \mathbb{R}^{n \times k}, H \in \mathcal{H}, L \in \mathbb{R}^k, g \in \mathbb{R}^n$. Then

$$F(M, H, L, g) := (U_1(M, H), U_2(M, H), L + c_2 M^+ g)$$

$$G(M, H, L, g) := -c_1 M L.$$

*Remark* 3.3. Vectors in $\mathbb{R}^m$ are considered to be column vectors. This means that they are treated as $m \times 1$ matrices in calculations.

The second, less abstract approach to RLLC describes the optimization process directly in a more conventional way.

**Require:**
- $\theta_0$ initial parameter vector
- $f(\theta)$: stochastic objective function with parameters $\theta \in \mathbb{R}^n$
- Two learning rates $c_1, c_2 > 0$
- Stability parameter $\epsilon$ for relaxed Penrose inverse
- $M_0 \in \mathbb{R}^{n \times k}$: initial memory units
- $L_0 \in \mathbb{R}^k$: initial learning law
- $H_0$: initial hidden state

$t := 0$: initialize time step

**while:** $\theta_t$ not converged **do**:

$$t \longleftarrow t + 1$$

$g_t := \nabla_\theta f_t(\theta_{t-1})$   Get New Gradient

$L_t := L_{t-1} + c_2 M_{t-1}^+ g_t$   RLLC Step

$(M_t, H_t) := U(M_{t-1}, H_{t-1}, g_t)$   Memory Update

$\theta_t := \theta_{t-1} - c_1 M_t L_t$   Learning Step

**Explanation of RLLC and remarks:**

The idea behind the learning rule update is that with the arrival of the new gradient $g_t$ the optimizer gains new (retrospective) information on how it could have done better in the previous learning step. Notice that the vector $M^+ g_t$ is the coefficient vector of the orthogonal projection of $g_t$ to the space spanned by the memory units when written as a linear combination of the memory units. This means that, if performed with the new law, the outcome of the parameter update in step $t - 1$ would have been $\theta_t - c_1 c_2 p_t$ instead of $\theta_t$ where $p_t$ is the orthogonal projection of $g_t$ to the space spanned by the memory units in the $t - 1$-th step. Notice that $(p_t, g_t) = (p_t, p_t) \geq 0$ and thus the change $-c_1 c_2 p_t$ points in a direction which improves the objective function.

The above heuristics does not take it into account that the objective function $f_t$ is also changing. This fact indicates that our update rule is more justified if the second learning rate $c_2$ is small and thus random effects have time to average out leaving only useful directions in the update. Also notice that the algorithm does not "go back in time" to perform the improved learning step. Instead it applies the

updated learning law with the updated memory units. This shows that the efficiency of the RLLC method depends on a type of consistency property. Roughly speaking it assumes that the notion of a "good learning law" does not change too much in time and so improvements of the past give improvements of the future. For this reason the choice of the memory update rule is a crucial issue which is one of the main topics of the second part of this paper.

*Remark* 3.4. The performance of the RLLC optimizer is dependent on the initialization of the learning law at the beginning. In practice it is not initialized to be $0$.

*Remark* 3.5. To avoid numerical instability, in practice we use a relaxed version of Penrose inverse which has a parameter $\epsilon$ set to a small number.

**Linear invariance of RLLC:** We close this chapter with a useful linear invariance property of the RLLC method. We will need the next two definitions.

**Definition 3.6.** Let $U$ be a memory update rule as above and let $Q \in \mathbb{R}^{k \times k}$ be an invertible matrix. We define the new memory update rule $U^Q$ in the following way: Let $M \in \mathbb{R}^{n \times k}$, $H \in \mathcal{H}$ and $U(MQ^{-1}, H, g) = (M_2, H_2)$. Then

$$U^Q(M, H, g) := (M_2 Q, H_2).$$

**Definition 3.7.** Two optimizers given by $(F, G)$ and $(F', G')$ with state spaces $\mathcal{S}$ and $\mathcal{S}'$ are called equivalent if there is a bijection $\phi : \mathcal{S} \to \mathcal{S}'$ (called an *isomorphism*) such that $\phi(F(S, v)) = F'(\phi(S), v)$ and $G(S, v) = G'(\phi(S), v)$. A *partial isomorphism* is a bijection between a subset of $\mathcal{S}$ and a subset of $\mathcal{S}'$ having the same property. If there is such a function we say that the two optimizers are partially equivalent on these two subsets. In particular, if two optimizers with memory $k$ states are partially equivalent on states with rank $k$ memory matrices then we call them *essentially equivalent*.

It is easy to see that if two optimizers are equivalent then they define the same optimization process if their initialization of internal states is isomorphic. If two optimizers are partially equivalent with partial isomorphism $\phi$ then the optimization processes are identical as long as they operate on states in the domain (and image) of $\phi$.

**Lemma 3.8** (Linear invariance of RLLC). *Let $U$ be a memory update rule as above and $Q \in \mathbb{R}^{k \times k}$ be an arbitrary matrix. Then the RLLC optimizer corresponding to $U$ is essentially equivalent to the RLLC optimizer corresponding to $U^Q$.*

*Proof.* We claim that the function $\phi(M, H, L) := (MQ, H, Q^{-1}L)$ is a partial isomorphism on states with rank $k$ memory matrices. This follows trivially from formulas in definition 3.2 and the fact that $(MQ)^+ = Q^{-1}M^+$ holds if $\mathrm{rank}(M) = k$. $\square$

## 4 LINEAR MEMORY UPDATES

Throughout this chapter we investigate linear memory update rules with no hidden states. Such an update rule is given by

$$U(M, g) := MB + ga^T \tag{1}$$

where $M \in \mathbb{R}^{n \times k}$ is the memory matrix, $g \in \mathbb{R}^n$ is a new gradient and $a \in \mathbb{R}^k, B \in \mathbb{R}^{k \times k}$ are fixed parameters of the update rule. In an optimization process this means that the memory unit $m_i$ represented by the $i$-th column of $M$ is updated to $a_i g + \sum_{j=1}^k B_{j,i} m_i$ when the new gradient $g$ is received.

**Linear memory optimizers with fixed learning law:** Linearly updated memory units are interesting independently of the RLLC method. We can directly obtain powerful optimizers by using a fixed hand designed learning law $L \in \mathbb{R}^k$. This type of optimizer, denoted by $\mathcal{L}(B, a, L)$ works by the equations:

$$M_t = M_{t-1}B + g_t a^T \ , \ \theta_t = \theta_{t-1} - M_t L.$$

If $k = 1$ then $B, a, L$ are single real numbers. The corresponding optimizer is a momentum SGD optimizer with decay parameter $B$ and learning rate $aL$. Another, important setting is described in the following lemma (for proof see Appendix A.1).

**Lemma 4.1.** *Let*

$$B = \begin{pmatrix} \beta & 0 \\ 0 & 0 \end{pmatrix}, \ a = \begin{pmatrix} 1 \\ 1 \end{pmatrix}, \ L = \begin{pmatrix} c\beta \\ c \end{pmatrix}$$

*Then the corresponding optimizer $\mathcal{L}(B, a, L)$ is Nestorv Accelerated Gradient with decay parameter $\beta$ and learning rate $c$.*

**Abstract rule of a memory unit:** There is a useful observation which sheds more light on what information linear memory stores if this memory update is iterated in an optimization process started with initial value $0^{n \times k}$ for $M$. By induction we have

$$U(...U(U(0^{n \times k}, g_1), g_2), ..., g_t) = \sum_{i=1}^{t} g_t a^T B^{t-i}.$$

We obtain that at time $t$ (after $t$ iteration of the update rule) the value of the $i$-th memory unit is given by

$$m_j = \sum_{i=0}^{t-1} g_{t-i} (a^T B^i)_j \tag{2}$$

where $(a^T B^i)_j$ denotes the $j$-th coordinate of the row vector $a^T B^i$. If we regard gradients with index 0 or negative index as 0 then the sum can be taken from 0 to infinity. Informally speaking, this means that $m_i$ is a fixed (time independent) linear combination of previous gradients going backwards in time. This linear combination is represented by the infinite sequence $\{(a^T B^i)_j\}_{i=0}^{\infty}$ for the $j$-th memory unit. We say that this infinite sequence is the *abstract rule* of the memory unit. To guarantee that older gradients are taken with decaying weight in (2) we need to assume that the spectral norm of $B$ is smaller than 1.

**Real momentum:** Memory updates in the case $k = 1$ are determined by two numbers: $\beta = B_{1,1}$ and $\alpha = a_1$. The update rule of the single memory unit $m$ in the $t$-th step is $m \longleftarrow \alpha g_t + \beta m$. (This is essentially the update rule of a momentum vector.) It follow from our formula that the abstract rule in this case is given by the geometric sequence $\beta^i \alpha$. In particular, in the $t$-th step we have that $m = \sum_{i=0}^{t} \alpha \beta^i g_{t-i}$.

**Complex momentum:** The case

$$B = \begin{pmatrix} \alpha & -\beta \\ \beta & \alpha \end{pmatrix} \tag{3}$$

has a distinguished role because such matrices represent complex numbers $\gamma = \alpha + \beta i$. In this special case the two memory units can be interpreted as the real and the complex parts of a single complex valued memory unit which describes a momentum vector with complex parameter $\gamma$. More precisely $m_1$ and $m_2$ are the real and the complex parts of a memory unit $m \in \mathbb{C}^n$ which is updated according to $m \longleftarrow g_t + \gamma m$.

**Jordan block of size** 2**:** Another interesting example for $k = 2$ is given by

$$B = \begin{pmatrix} \alpha & 1 \\ 0 & \alpha \end{pmatrix}$$

which is the so called Jordan block of size 2 with eigenvalue $\alpha$. The first one of the two memory units is the momentum vector of the gradients with parameter $\alpha$. However the second memory unit stores something new. It is the momentum vector of the first memory unit with parameter $a$. One can show that the abstract rule corresponding to this memory unit is given by the infinite sequence $0, 1, 2\alpha, 3\alpha^2, 4\alpha^3, \ldots$.

**Propagators and their unions:** In general, we call $k$ memory units $m_1, m_2, \ldots, m_k$ connected by a joint linear update rule a *propagator* of dimension $k$. Recall that such an object is described by a matrix $B \in \mathbb{R}^{k \times k}$ and a vector $a \in \mathbb{R}^k$. In the previous two examples each real number $\beta$ is associated with a propagator denoted by $M(\beta)$ of dimension 1 while complex numbers $\gamma$ are associated with a

propagator denoted by $CM(\gamma)$ of dimension 2. We call such propagators *momentum propagators*. It will be important for us that there is a simple operation on propagators that we call *union* and denote by $\oplus$. This is simply just taking the union of the corresponding memory units together and updating them independently. From a linear algebraic point of view, the matrix $B$ corresponding to the union of propagators is a block diagonal matrix whose blocks contain the matrices of the individual propagators. The vector $a$ corresponding to the union is the concatenation of the vectors of the propagators. Unions of momentum propagators will be called *multi momentum propagators*.

## 5 OPTIMIZERS WITH LINEAR MEMORY AND RLLC

In this chapter we discuss the basic properties of optimizers which combine linear memory and the RLLC method. We use the term LM-RLLC optimizers for them. Based on definition 3.2 and formula (1) one can produce the LM-RLLC optimizer $\mathcal{F}(B, a, c_1, c_2)$ with hyperparameters $B, a, c_1, c_2$. The corresponding update functions are given by

$$F(M, L, g) = (MB + ga^T, L + c_2 M^+ g)$$

$$G(M, L, g) = -c_1 ML$$

For the sake of completeness we describe the recursive optimization process.

**Definition 5.1.** (LM-RLLC optimization process) Let us fix the hyperparameters $B \in \mathbb{R}^{k \times k}, a \in \mathbb{R}^k, (c_1, c_2) \in \mathbb{R}^2$. Then the LM-RLLC optimizer with these hyperparameters is given by the equations

$$L_t = L_{t-1} + c_2 M_{t-1}^+ g_t$$

$$M_t = M_{t-1} B + g_t a^T$$

$$\theta_t = \theta_{t-1} - c_1 M_t L$$

where $M_0$ is the 0 matrix in $\mathbb{R}^{n \times k}$ and $L_0 \in \mathbb{R}^k$ is a suitable (typically non 0) vector.

Deeper mathematical analysis reveals that LM-RLLC optimizers can be transformed into a simpler, canonical form if we look at them up to equivalence. The key observation is a "basis independence" property of LM-RLLC optimizer functions.

**Theorem 5.2** (Basis independence of LM-RLLC optimizers)**.** *Let* $k \in \mathbb{N}, a \in \mathbb{R}^k, B \in \mathbb{R}^{k \times k}, (c_1, c_2) \in \mathbb{R}^2$ *and let* $Q \in \mathbb{R}^{k \times k}$ *be an invertible matrix. Then* $\mathcal{F}(B, a, c_1, c_2)$ *is essentially equivalent to* $\mathcal{F}(Q^{-1}BQ, Qa, c_1, c_2)$.

*Proof.* The optimizer $\mathcal{F}(B, a, c_1, c_2)$ is obtained from the linear memory update rule $U(M, g) = MB + ga^T$ with RLLC. Notice that $U^Q$ (in the sense of definition 3.6) is given by $U^G(M, g) = M(Q^{-1}BQ) + g(Qa)^T$. Then lemma 3.8 finishes the proof. $\square$

**Real Jordan normal form:** Theorem 5.2 together with a variant of the Jordan decomposition theorem implies that we can transform LM-RLLC optimizers into a very special form without changing the optimization process. The original form of the Jordan decomposition theorem says that if $B \in \mathbb{C}^{k \times k}$ is an arbitrary complex matrix then there is an invertible matrix $Q \in \mathbb{C}^{k \times k}$ such that $Q^{-1}BQ$ has a block diagonal form with each block being a so-called Jordan block. A Jordan block $J_m(\lambda)$ is a matrix of size $m \times m$ with $\lambda \in \mathbb{C}$ in the diagonal, 1 above the diagonal and 0 everywhere else. For example

$$J_3(\lambda) = \begin{pmatrix} \lambda & 1 & 0 \\ 0 & \lambda & 1 \\ 0 & 0 & \lambda \end{pmatrix}$$

There is a similar, although somewhat more complicated statement (called real Jordan normal form) if $B$ and $Q$ are required to be real matrices. In this case there are two types of blocks $J_m(\lambda)$ with $\lambda \in \mathbb{R}$ and $CJ_m(\alpha + \beta i)$ with $\alpha, \beta \in \mathbb{R}$. The second type of block has size $2m \times 2m$ and it "imitates" complex Jordan blocks with real matrices. This matrix is very similar to $J_m(\lambda)$ with the main difference being that each entry is replaced by a $2 \times 2$ matrix. The 0's and $1's$ are replaced

by 0 matrices and identity matrices. The $\lambda$ entries are replaced by the matrix in equation (3) which represents $\alpha + \beta i$ by a real matrix. For example

$$CJ_2(\alpha + \beta i) = \begin{pmatrix} \alpha & -\beta & 1 & 0 \\ \beta & \alpha & 0 & 1 \\ 0 & 0 & \alpha & -\beta \\ 0 & 0 & \beta & \alpha \end{pmatrix}$$

**Propagators of Jordan type:** Racall that an LM-RLLC optimizer is given by $B \in \mathbb{R}^{k \times k}, a \in \mathbb{R}^k$ and two learning rates. By transforming the matrix $B$ to its real Jordan normal form with a basis transformation given by $Q \in \mathbb{R}^{k \times k}$, we can divide the memory units into groups belonging to single blocks of type $J_m(\lambda)$ or $CJ_m(\alpha + \beta i)$. The block diagonal form of $Q^{-1}BQ$ guarantees that these groups do not interact with each other in memory updates and thus we can treat them as separate propagators. Recall that in this basis transformation considered in theorem 5.2 the vector $a$ transforms into $Qa$. By applying a statement which is slightly stronger then the Jordan decomposition theorem we can also guarantee that the part of $a$ in each block contains at most one coordinate with 1 and the rest is 0. We can also assume that this coordinate is the first one otherwise there are trivial memory units which store 0 in each step. By summarizing all of this we obtain propagators of very special type. Let $e_m \in \mathbb{R}^m$ denote the vector with 1 in the first coordinate and 0 in the rest. We denote the propagator corresponding to the pair $(J_m(\lambda), e_m)$ by $M_m(\lambda)$ and the propagator corresponding to $(CJ_m(\alpha + \beta i), e_{2m})$ by $CM_m(\alpha + \beta i)$. We call such propagators *Jordan block propagators*. If $m = 1$ then we omit the index and simply write $M(\lambda)$ and $CM(\alpha + \beta i)$. We obtain the next theorem.

**Theorem 5.3** (Normal forms of LM-RLLC optimizers). *Every LM-RLLC optimizer is essentially equivalent with another LM-RLLC optimizer where the memory update is of the form $P_1 \oplus P_2 \oplus \cdots \oplus P_r$ where each $P_i$ is a Jordan block propagator.*

By slightly abusing the notation we will also use the formula $P_1 \oplus P_2 \oplus \cdots \oplus P_r$ for the optimizer itself. For example $M(0.9) \oplus M_2(0.6) \oplus CM_2(0.3 + 0.2i)$ stands for a memory 7 optimizer where the memory units are grouped and updated according to the propagators $M(0.9)$, $M_2(0.6)$ and $CM_2(0.3 + 0.2i)$.

## 6 EXPERIMENTS

For our experiments, we used the Learned Optimization framework Metz et al. (2022a) as a starting point. The framework offers pre-trained and hyper parameter optimized optimizers. We compare our results with the most widely used optimizers as baseline: *Adam*, *SGD*, and *SGD with momentum*. We compare test loss and classification accuracy on MNISTDeng (2012), Fashin-MNISTXiao et al. (2017b), and CIFAR-10Krizhevsky (2009) datasets. We experimented with dense, convolutional and residual neural networks. The source code of our work is available publicly[1]. See implementation details in Appendix A.5.

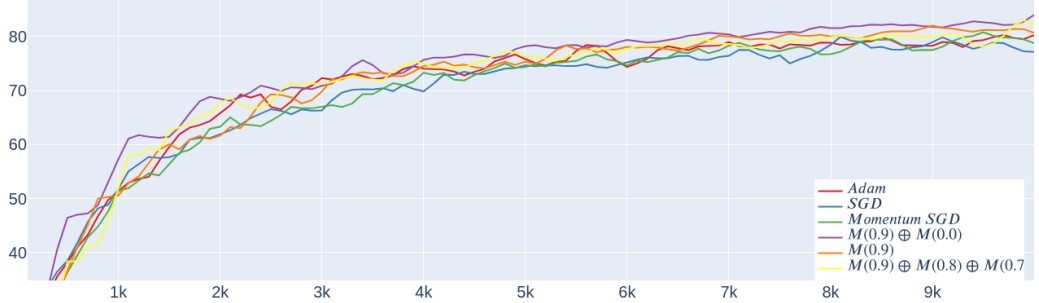

Figure 1: Test accuracy graphs of RLLC and benchmark optimizers, measured on the CIFAR-10 dataset, with the ResNet-20 network. RLLC optimizers show faster convergence and better generalization. See related plots and error bars in Appendix A.7.

---

[1]https://anonymous.4open.science/r/rllc-EB26/README.md

| | MNIST | | Fashion-MNIST | | CIFAR-10 | | | |
| --- | --- | --- | --- | --- | --- | --- | --- | --- |
| | MLP | Conv | MLP | Conv | MLP | Conv | ResNet-20 | |
| $SGD$ | 0.0882 | 0.0331 | 0.3426 | 0.3673 | 1.4084 | 0.8359 | 0.6010 | Loss |
| | 98.16 | 98.56 | 88.65 | 86.97 | 52.18 | 71.37 | 80.93 | Acc |
| $MomentumSGD$ | 0.0856 | 0.0324 | 0.3476 | 0.2732 | 1.4108 | 0.7850 | 0.5757 | Loss |
| | 98.22 | 98.97 | 88.67 | 90.78 | 51.99 | 73.17 | 81.36 | Acc |
| $Adam$ | **0.0758** | 0.0304 | 0.3407 | 0.2704 | 1.3858 | 0.7920 | 0.5857 | Loss |
| | 97.83 | 98.99 | 88.78 | 90.76 | 52.43 | 73.70 | 81.48 | Acc |
| $M(0.9)$ | 0.0844 | 0.0310 | 0.3408 | 0.2661 | 1.4021 | 0.8030 | 0.5301 | Loss |
| | 98.21 | 99.03 | 88.64 | 90.96 | 52.13 | 73.95 | 83.08 | Acc |
| $M(0.9)\oplus M(0.0)$ | 0.0888 | 0.0323 | 0.3475 | 0.2678 | 1.3973 | 0.7977 | 0.5353 | Loss |
| | **98.26** | 98.95 | 88.82 | 90.98 | 51.71 | 74.11 | **83.38** | Acc |
| $M(0.9)\oplus M(0.8)\oplus M(0.7)$ | 0.0829 | 0.0343 | 0.3359 | **0.2563** | 1.4142 | 0.7734 | **0.5268** | Loss |
| | 98.23 | 98.95 | 88.67 | 91.11 | 51.55 | 75.42 | 83.19 | Acc |
| $M_2(0.6)$ | 0.0801 | 0.0800 | **0.3220** | 0.4488 | **1.3444** | 1.0404 | 0.5811 | Loss |
| | 98.17 | 97.60 | 88.75 | 84.31 | **53.42** | 63.59 | 81.35 | Acc |
| $M(0.9)\oplus M_2(0.6)$ | 0.0861 | 0.0319 | 0.3536 | 0.2636 | 1.4155 | 0.7602 | 0.5354 | Loss |
| | 98.21 | 98.99 | 89.03 | 90.95 | 52.08 | **75.87** | 82.84 | Acc |
| $M(0.9)\oplus M(0.0)\oplus M_2(0.6)$ | 0.0877 | 0.0287 | 0.3498 | 0.2596 | 1.4028 | 0.7216 | 0.5393 | Loss |
| | 98.23 | **99.03** | 89.14 | **91.29** | 51.83 | 75.76 | 82.73 | Acc |
| $M_3(0.6)$ | 0.0797 | 0.0539 | 0.3282 | 0.3735 | 1.3798 | 0.9433 | 0.5445 | Loss |
| | 98.22 | 98.31 | **89.25** | 87.08 | 53.37 | 66.68 | 82.43 | Acc |
| $M(0.9)\oplus M(-0.9)\oplus CM(0.9i)$ | 0.0873 | 0.0334 | 0.3671 | 0.2624 | 1.4029 | 0.7647 | 0.5337 | Loss |
| | 98.09 | 98.97 | 88.57 | 91.06 | 52.10 | 75.44 | 83.07 | Acc |

Table 1: Loss and accuracy are reported across three different datasets, using three distinct network architectures. The first three rows are dedicated to benchmark optimizers, whereas the subsequent rows showcase our results. The best benchmark result for each task (dataset and architecture pair) are highlighted in blue. Instances where our optimizer exceeds the best baseline result are marked in green. Additionally, the absolute best value for each task is emphasized in bold font. The results represent the average of three runs with different random seeds. Standard deviation values are provided in Appendix A.4.1, demonstrating consistent performance across all runs.

**RLLC based adaptive learning rate:** One of the simplest case of the RLLC method is already interesting. If there is a single memory unit containing the momentum of previous gradients then RLLC yields an adaptive version of the momentum SGD optimizer. In this case the learning law contains a single coefficient, that defines an adaptively changing learning rate for the momentum SGD. Our experiments show that this upgrade outperforms the plain momentum SGD method, showcasing the power of RLLC. See $M(0.9)$ results in Table 1 and on Figure 1. Note that RLLC can be applied to an arbitrary optimizer by introducing a single memory unit storing the last learning step. In a similar way we obtain a version of the optimizer with an adaptive learning rate. However it may depend on the optimizer whether it leads to a performance boost or not.

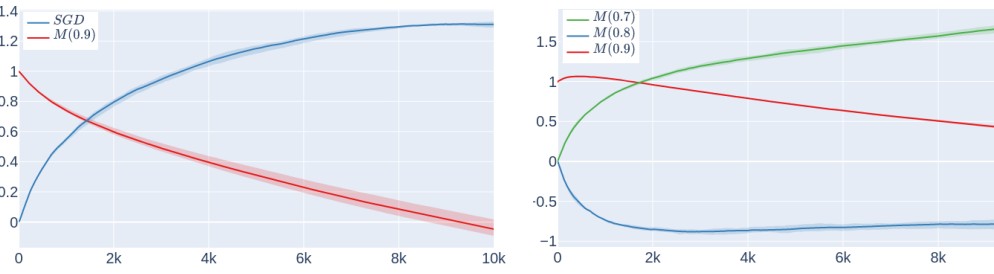

(a) The figure shows the $M(0.9)\oplus M(0.0)$ optimizer's transition between momentum SGD and SGD, briefly aligning with the NAG optimizer around the 2k step.

(b) The figure shows an interesting negative coupling between $M(0.8)$ and $M(0.7)$. See further details in Appendix A.3.

Figure 2: Analysis of the memory unit coefficients over time for different optimizers: $M(0.9)\oplus M(0.0)$ (left) and $M(0.9)\oplus M(0.8)\oplus M(0.7)$ (right).

**Mixing SGD and momentum SGD:** We observe an intriguing phenomenon when we enhance the memory unit of the previous method with the current gradient and monitor the learning law throughout the training process. As shown in Figure 2a, during the initial phase of training, the coefficient of the $M(0.9)$ memory unit is predominant. However, as training advances, the coefficient of the $M(0)$ unit increases, leading to a reversal in the significance of the two memory units. Our experiment supports Keskar & Socher (2017a) findings. Table 1 and Figure 1 show results for $M(0.9) \oplus M(0)$. An interesting additional detail is that in between the two extremal phases there exists a phase which emulates the Nesterov Accelerated Gradient (NAG) method. This occurs when the coefficient of the Momentum SGD memory unit, divided by the coefficient of the SGD unit, equals the decay parameter of the momentum SGD. (For more details, see Appendix A.2).

**Multi-momentum propagators:** In our experiment we investigated optimizers of the form $M(\beta_1) \oplus M(\beta_2) \oplus \cdots \oplus M(\beta_k)$. We have not optimized the hyperparameters $\beta_i$ but we found very promising settings with a few trials. Our results are therefore illustrative and the fine tuning (depending on the type of network) is subject to further research. In Table 1 we describe our experiments with $M(0.9) \oplus M(0), M(0.9) \oplus M(0.6), M(0.9) \oplus M(0.8) \oplus M(0.7)$ and $M(0.9) \oplus M(0.6) \oplus M(0)$ optimizers. Quite surprisingly the simplest one $M(0.9) \oplus M(0)$ (which is mentioned earlier) is the most reliable. However on certain tasks it is outperformed by the memory 3 settings. Figure 2b illustrates an interesting coupling between the coefficient of $M(0.8)$ and $M(0.7)$ memory units. See further details in Appendix A.3.

$M_m(\lambda)$ **propagator for** $m \geq 2$**:** Jordan block propagators of the form $M_m(\beta)$ and $CM_m(\beta)$ with $m \geq 2$ are easy to implement in our code. In our experiments we focused on type $M_m(\beta)$ propagators with $m = 2, 3$. It is also interesting to combine them with other propagators. Table 1 shows results for $M_2(0.6), M_3(0.6)$ and $M(0.9) \oplus M_2(0.6)$. These configurations surpass the baseline optimizers in many tasks and also surpass pure multi-momentum propagators in some specific tasks.

**Complex-moment propagators:** Another interesting possibility in our framework is the usage of complex-momentum propagators. One particular example that we experimented with is the case of $M(\beta) \oplus M(-\beta) \oplus CM(\beta i)$. This choice in not random. It comes from the Jordan normal form of the permutation matrix corresponding to the cyclic shift on $4$ elements multiplied with $\beta$. This particular propagator is closely related to Fourier analysis.

# 7 LIMITATIONS

Using memory units comes at a cost. Each memory unit is a vector in the parameter space $\mathbb{R}^n$. In our experiments, we opted for relatively small or medium-sized architectures. However, for architectures with a vast parameter space, our approach with many memory units could prove to be too memory-intensive.

It's also worth noting that our experiments were conducted on relatively small datasets, and future work should explore experiments on larger datasets.

# 8 CONCLUSION

Our experiments demonstrate that the RLLC method is capable of boosting the performance of classical optimizers (such as SGD and momentum SGD) by combining them and making them more adaptive. Furthermore the case of linearly updated memory units provides a mathematically elegant framework with many new types of promising optimizers such as the ones corresponding to larger Jordan blocks, complex numbers and their combinations. We regard this paper as a starting point for future research in the frame of which the full potential of our approach is explored. One possible research direction is to introduce adaptively changing memory update rules. In particular, in the linear setting the pair $B \in \mathbb{R}^{k \times k}, a \in \mathbb{R}^k$ (see Section 4) is fixed for the whole optimization process in the current version. It would be interesting to study a version were $B$ and $a$ are also adaptively changing throughout learning.

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

# A   APPENDIX

## A.1   NESTEROV ACCELERATED GRADIENT (NAG) AS A MEMORY 2 OPTIMIZER

We claim that a basic version of the NAG optimizer with decay parameter $\beta$ and learning rate $c$ is equivalent with the linear memory optimizer with two memory units $M(\beta)$, $M(0)$ and fixed learning law $(c\beta, c)$. This is a special LM-RLLC optimizer where the learning rate is 1 and the meta learning rate is 0 and thus the learning law does not change. To verify this claim, recall that the NAG optimizer is given by the iteration of the following steps:

$$\phi_{t+1} = \theta_t - c\nabla f(\theta_t)$$
$$\theta_{t+1} = \phi_{t+1} + \beta(\phi_{t+1} - \phi_t).$$

where $\phi_1 = \theta_1$ is the initial parameter vector and $f$ is the objective function to be minimized. Let us introduce

$$v_{t+1} := \phi_t - \phi_{t+1}.$$

With this notation we have the update rules:

$$v_{t+1} = \beta v_t + c\nabla f(\theta_t),$$
$$\theta_{t+1} = \theta_t - (c\nabla f(\theta_t) + \beta v_{t+1}).$$

Observe that if we introduce the update rule

$$m_{t+1} := \beta m_t + \nabla f(\theta_t)$$

then $v_t = r m_t$ holds at any given time $t$. Furthermore $m_t$ corresponds to the propagator $M(\beta)$. With this notation we have that

$$\theta_{t+1} = \theta_t - (c\nabla f(\theta_t) + c\beta m_{t+1}).$$

This verifies our claim since $\nabla f(\theta_t)$ is the propagator $M(0)$.

A possible source of confusion is that NAG is known in two slightly different but interrelated versions, depending on which of the two sequences, $\theta_t$ and $\phi_t$, is considered the actual learning step (while the other is regarded as an auxiliary step).

In one version, the points $\theta_i$ are considered "look-ahead points" for computing the gradients, while $\phi_i$ are viewed as the learning steps. Notice that in this version, the gradients are NOT computed at the learning steps; they are calculated at the look-ahead points.

The sequence $\theta_i$ of the look-ahead points is closely coupled with the learning steps, and their difference is a single SGD step, which diminishes over time. This means that they converge to the same point in the parameter space. For this reason, one can view NAG in a different way, where the learning steps are the look-ahead points $\theta_t$, and this is often done in the literature. This second version of NAG is more natural for our framework because the gradients are computed at the actual learning steps.

To illustrate this we compute the first few NAG steps. To start with the iteration we introduce initial values by $\phi_1 = \theta_1$. We will use the short hand notation $g_t = \nabla(\theta_t)$. With this notation, the recursion takes the form

$$\phi_{t+1} = \theta_t - cg_t \; , \; \theta_{t+1} = \phi_{t+1} + \beta(\phi_{t+1} - \phi_t).$$

Now we have the following:

$$\phi_2 = \theta_1 - cg_1$$
$$\theta_2 = \theta_1 - c(1 + \beta)g_1$$
$$\phi_3 = \theta_1 - c(1 + \beta)g_1 - cg_2$$
$$\theta_3 = \theta_1 - c(1 + \beta + \beta^2)g_1 - c(1 + \beta)g_2$$
$$\phi_4 = \theta_1 - c(1 + \beta + \beta^2)g_1 - c(1 + \beta)g_2 - cg_3$$
$$\theta_4 = \theta_1 - c(1 + \beta + \beta^2 + \beta^3)g_1 - c(1 + \beta + \beta^2)g_2 - c(1 + \beta)g_3$$

Observe that the sequence $\phi_i$ is the sequence given by a more classical version of NAG while $\theta_i$ is given by the memory two $M(0), M(\beta)$ optimizer as described above. This explains the exact relationship between the two sequences. Notice that $\phi_{t+1} - \theta_t = -cg_t$ where $-cg_t$ is a single gradient step which converges to 0 as the optimization gets closer to the optimum.

## A.2 RLLC INTERPOLATIONS BETWEEN SGD, MOMENTUM SGD AND NESTEROV ACCELERATED GRADIENT (NAG)

As we have already explained, momentum SGD method in our interpretation is represented as the propagator $M(\beta)$ where $0 \leq \beta < 1$ is the decay parameter. In particular $M(0)$ corresponds to a memory unit which stores the last gradient seen by the optimizer. In this sense, a memory 1 optimizer with memory unit $M(0)$ is basically an SGD optimizer. The learning law in this case is a single real number which manifests as a learning rate.

If an optimizer has two memory units $M(\beta)$ and $M(0)$ then by changing the learning law (described by a pair of real numbers) we can continuously interpolate between momentum SGD, pure SGD and NAG. In the next list we summarize the meanings of special learning laws for $M(\beta), M(0)$.

1. $(0, c)$ : SGD

2. $(c, 0)$ : momentum SGD with decay parameter $\beta$

3. $(c\beta, c)$ : NAG with decay parameter $\beta$

where the learning rate is $c * \mathrm{lr}$. Notice that since $\beta$ is a prescribed fix number, the above three cases don't cover all possible learning laws for the pair $M(\beta), M(0)$. This enables the LM-RLLC optimizer with memory $M(\beta), M(0)$ to find interesting interpolations between these three classical optimizers.

## A.3 ADDITIONAL MATHEMATICAL OBSERVATIONS

**On the memory units of $M_k(\lambda)$ propagators:** The $M_k(\lambda)$ propagator can naturally be interpreted as an iterated momentum propagator. Let $m_1, m_2, \ldots, m_k$ denote the the memory units. The update rule of $M_k(\lambda)$ is given by

$$m_1 \leftarrow m_1\beta + g$$

$$m_i \leftarrow m_i\beta + m_{i-1} \text{ for } i \geq 2.$$

Thus $m_1$ is the momentum vector of the gradient and $m_i$ (for $i \geq 2$) is the momentum vector of $m_{i-1}$. One can compute that the abstract rule of $m_i$ is given by the sequence $\{\beta^{j-i+1}\binom{j}{i-1}\}_{j=0}^{\infty}$. It follows that the subspace generated by the abstract rules of the memory units is the space of all sequences of the form $\{p(j)\beta^j\}_{j=0}^{\infty}$ where $p$ is a polynomial of degree at most $k-1$. This means that we can associate such a polynomial with each learning law. The RLLC method for $M_k(\lambda)$ basically adaptively navigates in this polynomial space.

**Relation between Multi-momentum and $M_k(\lambda)$ propagators:** The progression of the learning law of $M(0.9) \oplus M(0.8) \oplus M(0.7)$ presents an interesting phenomenon. As Figure 2b shows, the coefficients of $M(0.8)$ and $M(0.7)$ memory units are noticeably coupled with opposite sign. One might assume that the algorithm is just trying to cancel their effects, but the performance improvement compared to $M(0.9)$ suggests that something more interesting is happening here. A deeper explanation relates this optimizer to another one of the form $M(0.9) \oplus M_2(0.75)$. More precisely if we consider $\mathcal{O}(\epsilon) = M(0.9) \oplus M(0.75 + \epsilon) \oplus M(0.75 - \epsilon)$ we find that as $\epsilon$ goes to 0 the subspace spanned by the abstract rules of the memory units converges to the subspace spanned by the memory units of $M(0.9) \oplus M_2(0.75)$. In theory this convergence means that the optimizers themselves converge. Note that in practice we can not model $M(0.9) \oplus M_2(0.75)$ by $\mathcal{O}(\epsilon)$ because numerical instability arises if $\epsilon$ is very small.

## A.4 ADDITIONAL EXPERIMENTS

### A.4.1 RESULTS CONSISTENCY

To address concerns about the variability of our results, we computed and included the standard deviation values for each experiment in Table 2. The standard deviation values indicate that the observed improvements are consistent across different seeds, with the standard deviation being substantially smaller than the performance gains over baseline methods.

### A.4.2 OPTIMIZER TRAINING COST

We compared our RLLC optimizers' performance with the benchmark optimizers (implemented in the Optax library). In this experiment we skipped all evaluation and logging features of the training process, and we only measured, how much time it takes the optimizer to reach the appointed iteration step. RLLC optimizers' compute time is approximatively the same as the benchmark optimizers (both lock time and in FLOPs). With careful code optimization, performance of the benchmark optimizers (with the same number of memory units) is achievable.

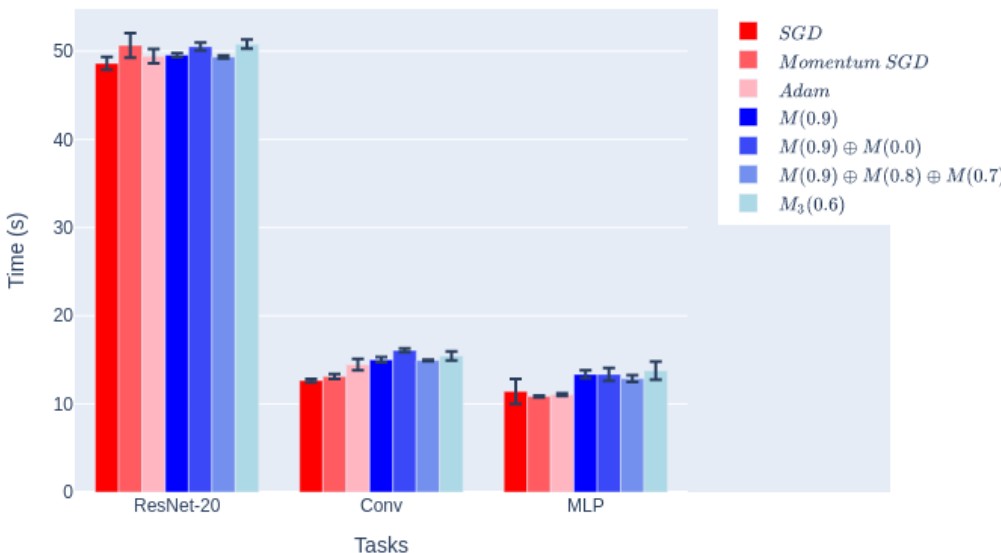

Figure 3: Performance comparison of RLLC and benchmark optimizers, measured on the CIFAR-10 dataset, with an MLP, convolutional, and ResNet-20 network.

## A.5 IMPLEMENTATION DETAILS

### A.5.1 NETWORK ARCHITECTURES AND TRAINING DETAILS

**Dense network** Our dense network comprises three hidden layers, each with a width of 128 and followed by a ReLU activation function. We did not include any normalization layers.

**Convolutional network** Our convolutional network features a depth of three, with channel widths of 32, 64, and 64, each followed by a ReLU activation function. We did not incorporate any normalization layers. Following the convolutional layers, we apply max pooling and then a final dense layer.

**ResNet-20** Our ResNet-20 variant adheres to established conventions for CIFAR-10, employing a three-level architecture with three residual blocks at each level. Each residual block is composed of the following sequence of layers: Convolution-Batch Normalization-ReLU-Convolution-Batch Normalization. A ReLU operation is applied after the addition operation in each residual block. The convolution kernels are 3x3 in size.

| | MNIST | | Fashion-MNIST | | CIFAR-10 | | | |
| --- | --- | --- | --- | --- | --- | --- | --- | --- |
| | **MLP** | **Conv** | **MLP** | **Conv** | **MLP** | **Conv** | **ResNet-20** | |
| $SGD$ | 98.1573 0.03 | 98.5628 0.09 | 88.6537 0.05 | 86.9660 0.38 | 52.1822 0.45 | 71.3706 1.08 | 80.9269 0.27 | Acc Std |
| $MomentumSGD$ | 98.2232 0.05 | 98.9748 0.10 | 88.6735 0.15 | 90.7799 0.33 | 51.9877 0.16 | 73.1672 0.58 | 81.3555 0.55 | Acc Std |
| $Adam$ | 97.8343 0.10 | 98.9880 0.04 | 88.7757 0.01 | 90.7602 0.16 | 52.4295 0.58 | 73.7045 0.29 | 81.4808 1.05 | Acc Std |
| $M(0.9)$ | 98.2133 0.07 | 99.0341 0.05 | 88.6439 0.16 | 90.9579 0.24 | 52.1328 0.30 | 73.9517 1.20 | 83.0795 0.66 | Acc Std |
| $M(0.9){\oplus}M(0.0)$ | **98.2595 0.15** | 98.9452 0.04 | 88.8219 0.18 | 90.9843 0.17 | 51.7075 0.63 | 74.1067 0.34 | **83.3762 0.78** | Acc Std |
| $M(0.9){\oplus}M(0.8){\oplus}M(0.7)$ | 98.2331 0.08 | 98.9484 0.10 | 88.6702 0.16 | 91.1096 0.11 | 51.5526 0.16 | 75.4153 0.41 | 83.1916 0.76 | Acc Std |
| $M_2(0.6)$ | 98.1738 0.05 | 97.5969 0.14 | 88.7460 0.41 | 84.3091 0.07 | **53.4184 0.19** | 63.5878 0.20 | 81.3489 0.77 | Acc Std |
| $M(0.9){\oplus}M_2(0.6)$ | 98.2133 0.04 | 98.9880 0.04 | 89.0295 0.13 | 90.9481 0.07 | 52.0800 0.34 | **75.8736 0.13** | 82.8422 0.24 | Acc Std |
| $M(0.9){\oplus}M(0.0){\oplus}M_2(0.6)$ | 98.2298 0.08 | **99.0342 0.12** | 89.1350 0.24 | **91.2942 0.23** | 51.8295 0.44 | 75.7648 0.28 | 82.7334 0.81 | Acc Std |
| $M_3(0.6)$ | 98.2199 0.04 | 98.3089 0.16 | **89.2537 0.18** | 87.0847 0.16 | 53.3656 0.75 | 66.6766 0.49 | 82.4301 0.27 | Acc Std |
| $M(0.9){\oplus}M(-0.9){\oplus}CM(0.9i)$ | 98.0947 0.11 | 98.9748 0.05 | 88.5713 0.13 | 91.0568 0.17 | 52.0965 0.69 | 75.4417 0.24 | 83.0663 0.89 | Acc Std |

Table 2: Accuracy and standard deviation values are reported across three different datasets, using three distinct network architectures. The first three rows are dedicated to benchmark optimizers, whereas the subsequent rows showcase our results. The best benchmark result for each task (dataset and architecture pair) are highlighted in blue. Instances where our optimizer exceeds the best baseline result are marked in green. Additionally, the absolute best value for each task is emphasized in bold font. The results are the average of 3 runs with different random seeds.

### A.5.2 TRAINING DETAILS, HYPERPARAMETERS

In all reported experiments, we employed a batch size of 128 and trained the models for 10,000 iterations. We did not use a learning rate scheduler, to avoid any potential variance in its effect across different optimizers. We run every experiment with 3 different seed, and reported the average of the results.

During our hyperparameter optimization process, we tested the following potential values:

- **Benchmark optimizers**

    learning rate: 1e-7, 3e-7, 1e-6, 3e-6, 1e-5, 3e-5, 1e-4, 3e-4, 1e-3, 3e-3, 1e-2, 3e-2, 1e-1, 3e-1, 1

- **Our optimizers**

    learning rate: 0.001, 0.003, 0.01, 0.03, 0.1, 0.3

    learning law - learning rate: 0.003, 0.01, 0.03

### A.5.3 TRAININD DATASETS

**CIFAR-10** The CIFAR-10 dataset Krizhevsky et al. consists of 60000 $32x32$ colour images in 10 classes, with 6000 images per class. There are 50000 training images, and 10000 test images. We used the canonical train–validation-test split, with 45000 train, 5000 validation, and 10000 test images. As a preprocessing, we normalized the images with the means $(0.4914, 0.4822, 0.4465)$ and standard deviations $(0.2023, 0.1994, 0.2010)$ for the three RGB channels, respectively. On the *ResNet* task we used random resized crop (with zoom scale 0.8-1.2), horizontal flip, and random rotation.

**Fashion-MNIST** The Fashion-MNIST dataset Xiao et al. (2017a) consists of $70000$ $28x28$ monochrom images in 10 classes, with 7000 images per class. There are 60000 training images, and 10000 test images. We used the canonical train–validation-test split, with 54000 train, 6000 validation, and 10000 test images. As a preprocessing, we normalized the images with the mean 0.3 and stadard deviation 0.3.

**MNIST** The MNIST dataset LeCun et al. (2010) consists of $60000$ $28x28$ monocrom images in 10 classes, with 7000 images per class. There are 60000 training images, and 10000 test images. We used the canonical train–validation-test split, with 54000 train, 6000 validation, and 10000 test images. As a preprocessing, we normalized the images with the mean 0.1307 and standard deviation 0.3081.

## A.6 COMPUTATIONAL RESOURCES

For our experiments we used a server with 8 A10040GB GPUs. We reported outcomes from a total of 16 optimizers, each optimized for hyperparameters across seven distinct tasks (architecture - dataset pair). One round of hyperparameter optimization with three different random seeds took approximately 4 hours on our server, and we were able to run 16 paralelly.

Therefore, all of our results can be replicated in about 28 hours using the same setup, or in 224 hours on a single A100 40GB GPU.

## A.7 SUPPLEMENTARY PLOTS

Figure 4 shows additional accuracy plots for MLP and Convolutional tasks. Figure 5 demonstrates, that the test accuracy is consistent on different random seeds for the demonstrated experiments.

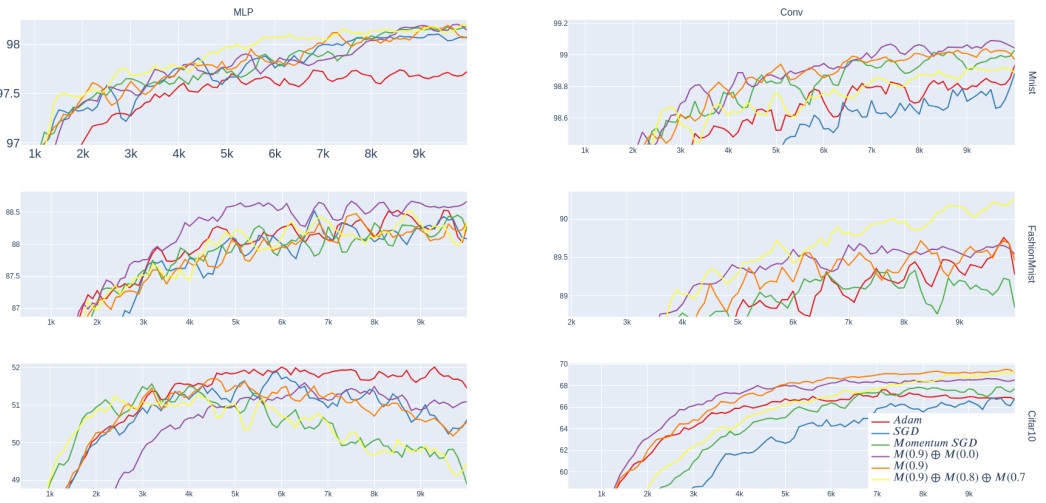

Figure 4: Test accuracy graph of some RLLC optimizer, comparing with benchmark optimizers. On most of the tasks RLLC optimizers perform better, than the benchmark optimizers.

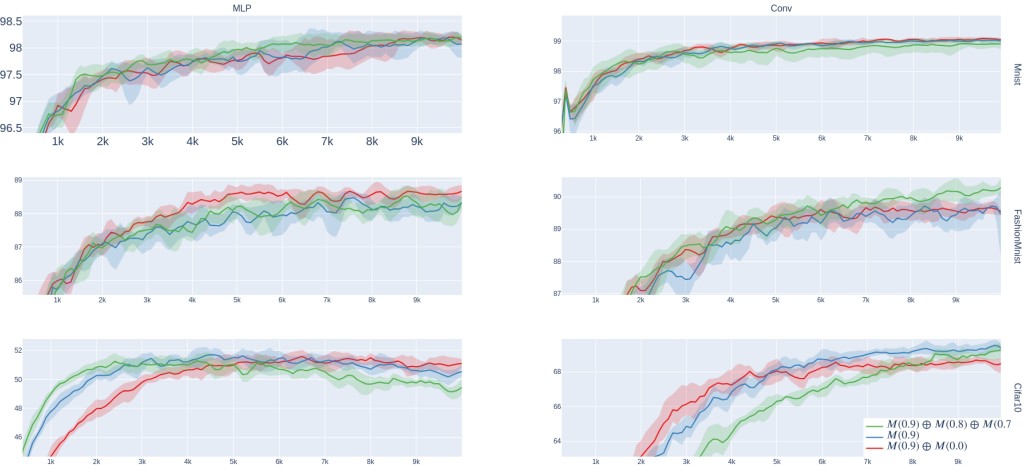

Figure 5: Test accuracy graph of some RLLC optimizer, with min-max interval, trained from 3 random seed initialization. The accuracy does not vary a lot, suggesting, that RLLC is robust.

