# OpenReview forum: "Dynamic Memory Based Adaptive Optimization"
_ICLR.cc/2025/Conference — Submitted to ICLR 2025_

### Official Review · Reviewer_fhjC · 2024-11-02

**Soundness:** 2
**Presentation:** 1
**Contribution:** 2
**Rating:** 3
**Confidence:** 2

**Summary:**

The authors introduce a novel framework, Retrospective Learning Law Correction (RLLC), which leverages multiple memory units to optimize learning processes in neural networks. By defining optimizers based on the number of memory vectors they store, the paper categorizes classical optimizers (SGD, momentum SGD, Adam) and posits that utilizing additional memory can lead to improved performance.

**Strengths:**

The core idea of the paper is to extend this memory concept by developing optimizers that can leverage multiple memory units (beyond just one or two), leading to potentially improved performance. To address the questions of what to store and how to utilize these stored units effectively, the authors propose the RLLC method. RLLC introduces a dynamically varying linear combination, or "learning law," that determines how memory units interact and contribute to the update at each learning step.

**Weaknesses:**

- **Experiments**: The paper lacks large-scale experiments, relying only on MNIST and CIFAR datasets, which are relatively outdated benchmarks. Even on these small-scale datasets, the performance improvements are modest. Evaluating RLLC on large language model (LLM) pretraining tasks would strengthen the results.
- **Convergence Analysis**: The paper does not provide a convergence analysis for RLLC, leaving theoretical insights unaddressed.
- **Memory Requirements**: While RLLC aims to improve performance, it requires storing multiple memory units and, in practice, demands more memory units than Adam to achieve optimal results.

**Questions:**

Unlike Adam, RLLC does not rely on second-order momentum; instead, it uses a linear combination of gradients. However, many other works have explored multi-momentum optimizers, which require additional memory to store multiple momentum terms.

---

### Official Review · Reviewer_24vq · 2024-11-03

**Soundness:** 1
**Presentation:** 1
**Contribution:** 2
**Rating:** 3
**Confidence:** 4

**Summary:**

This paper proposes a general gradient descent framework for retrieving retrospective information from new gradients by multiple memory units. When specialized with linear update rules, the proposed framework recovers classic optimizers with one memory unit such as SGD and NAS, and can be further decomposed to Jordan blocks. This paper then compares optimizers generated from Jordan blocks to classic optimizers across various datasets and neural networks, and demonstrates a competitive competence. This paper also provides certain theoretical explorations.

**Strengths:**

(S1) The proposed framework can be decomposed into Jordan blocks when constrained to linear memory, providing a mathematical foundation for developing and studying such ‘linear’ optimizers.

**Weaknesses:**

Overall, this paper is not in proper writing and structure for publication, and there are certain flaws in both the theoretical analysis and experiments that might invalidate the whole paper. I will consider raising the score if concerns are addressed.

(W1) The citation format is unclear (misuse of \citet and \citep across the whole paper, missing space before many citations).

(W2) Some undefined notations, e.g., U_1 and U_2 at line 175, and typos exist, e.g. U_G at line 360.

(W3) Experiments lack statistical significance: results are averaged from only three different seeds.

(W4) Lemma 3.8 might be flawed or has a typo because the proof uses Q^{-1} where Q is an arbitrary k*k matrix in its statement.

(W5) The paper is hard to read because of the mismatch between its linear focus and the opening abstract framework, I thus suggest putting the abstract framework to the end or to the appendix.

**Questions:**

(Q1) I am curious about your ‘consistency property’ at line 218, could you provide some examples?

(Q2) This paper aims to improve past updates by extracting retrospective information from current gradients. Instead of using crafted memory units, have you thought about using parameterized distributions to extract? Some evolution strategies papers might be of help, such as [1](a summarized one) and [2](has parts similar to the equation at line 285).

[1]Ollivier, Y., Arnold, L., Auger, A., & Hansen, N. (2017). Information-geometric optimization algorithms: A unifying picture via invariance principles. Journal of Machine Learning Research, 18(18), 1-65.
[2]Zhang, Z., Wei, Y., & Sui, Y. (2024). An Invariant Information Geometric Method for High-Dimensional Online Optimization. Proceedings of the 6th Annual Learning for Dynamics & Control Conference, PMLR 242:641-653.

---

### Official Review · Reviewer_e6Tz · 2024-11-04

**Soundness:** 3
**Presentation:** 3
**Contribution:** 2
**Rating:** 3
**Confidence:** 3

**Summary:**

A new optimizer called RLLC is proposed. This method is designed to calculate a dynamically varying linear combination of memory units (i.e., historical trajectory vectors in the parameter space) for the update of the parameter vector $\theta$.

**Strengths:**

1. The paper is easy to follow, and the authors accordingly explain the idea and core contribution of RLLC.

2. The connection between RLLC and other adaptive optimizers is direct, which may motivate more work in this direction.

**Weaknesses:**

1. The new optimizer should be verified on more challenging problems, e.g., LLMs or Diffusion Models.

2. For architectures with a vast parameter space (like LLMs), RLLC with many memory units could prove to be too memory-intensive, setting a large obstacle to extending this method to more applications.

**Questions:**

More theory discussion on what information should be stored in memory units may be necessary.

---

### Official Review · Reviewer_xsY7 · 2024-11-07

**Soundness:** 3
**Presentation:** 3
**Contribution:** 2
**Rating:** 5
**Confidence:** 2

**Summary:**

The paper introduces a novel framework for first-order optimization called “Retrospective Learning Law Correction” (RLLC), which adapts optimizers by dynamic adjusting the update coefficients for memory units, e.g., momentum vectors. Unlike traditional optimizers such as SGD or Adam, where these coefficients are fixed, the RLLC approach utilizes these memory units to adjust learning laws dynamically. The authors present a mathematical framework where optimizers can store and update memory units (up to four in the examples), and show how popular optimizers such as SGD, momentum SGD, and NAG can be examined using the new framework as cases of “fixed” memory units. The latter comparison suggests that a more flexible/adaptable mechanism may be more powerful. The paper includes numerical experiments where heuristically tuned case of RLLC of up to four memory units are compared against SGD, momentum, and Adam. Some variants of RLLC are found to outperform these existing methods on certain cases.

**Strengths:**

- Although I am relatively unfamiliar with the literature in this area, the introduction of the RLLC framework appears to provide a novel way to study gradient-based methods for model training. Furthermore, the generalization of memory units provides a method to adapt learning rates using retrospective gradient analysis, which could be a significant advancement in optimization techniques. To my knowledge, this is an interesting new direction, as most conventional optimizers used fixed update rules.
- The mathematics (primarily linear algebra) of the paper are presented well. The authors include a detailed theoretical analysis, including proofs of equivalence in certain cases to existing optimization methods. Furthermore, the authors study Jordan block decompositions of the proposed update rules to improve practical implementations.
- In addition to the new optimizers, the paper essential provides a theoretical framework to analyze and compare existing optimizers (e.g., SGD, momentum-based methods), and furthermore make non-trivial modifications by allowing the dynamic adjustment of memory units. Since existing methods correspond to fixed-coefficient cases, adapting these can enable interpolation between well-known strategies where they work well respectively.

**Weaknesses:**

- The main weaknesses of this paper relate to its practical relevance to the machine learning community at large.
  - Firstly, the proposed RLLC method is more computationally and memory intensive than existing frameworks, and the authors note that they only test small-to-medium scale networks due to this fact. It would be interesting to see how RLLC can help the training of more modern architectures than fully connected and convolutional networks. Similarly, the experiments are limited to small datasets, e.g., MNIST, FashionMNIST.
  - Despite the above, the performance benefits of RLLC are relatively unclear. For many of the included cases, only certain cases of RLLC, both in terms of number of memory blocks and tunable hyperparameters, outperform the existing, cheaper methods.
  - Related to the above, the RLLC method introduces a significant number of hyperparameters to tune (the number of memory blocks and the propagator coefficients). The authors do not investigate their tuning or provide strategies for their selection despite their importance (see above comment).

**Questions:**

- Can the computational and memory complexities of various orders of RLLC be exactly quantified and compared against existing methods?
- Given the nice comparisons of equivalence points in Figure 2, can the best- and/or worst-case improvements for various orders of RLLC be studied if some assumptions are made about the training setting? E.g., when may some optimizer be expected to outperform another in this framework?

---

### Meta-Review · Area_Chair_F7wC · 2024-12-07

**Metareview:**

The paper considers the stochastic first-order methods for standard stochastic optimization problems. An Retrospective Learning Law Correction (RLLC) framework has been proposed. This framework not only contains multiple popular existing algorithms, it has the ability to construct new first-order algorithms. Preliminary experiments has been conducted to validate the performance of the proposed framework. This paper has not analyzed the convergence property of the proposed algorithm, and hence should be viewed as a heuristic research. However, as a heuristic research paper, as pointed out by several reviewers, the numerical validation is only on small scale problems, and lacks essential experimental results on harder problems.

Finally, because the authors have yet replied any of the reviewers' comment during the rebuttal period, we decide to reject this paper.

**Additional Comments On Reviewer Discussion:**

No rebuttal from the authors.

---

### Decision · Program_Chairs · 2025-01-22

Reject